# Exploration of the Possible Relationships Between Gut and Hypothalamic Inflammation and Allopregnanolone: Preclinical Findings in a Post-Finasteride Rat Model

**DOI:** 10.3390/biom15071044

**Published:** 2025-07-18

**Authors:** Silvia Diviccaro, Roberto Oleari, Federica Amoruso, Fabrizio Fontana, Lucia Cioffi, Gabriela Chrostek, Vera Abenante, Jacopo Troisi, Anna Cariboni, Silvia Giatti, Roberto Cosimo Melcangi

**Affiliations:** 1Department of Pharmacological and Biomolecular Sciences “Rodolfo Paoletti”, Università degli Studi di Milano, Via Balzaretti 9, 20133 Milano, Italy; silvia.diviccaro@unimi.it (S.D.); roberto.oleari@unimi.it (R.O.); federica.amoruso@unimi.it (F.A.); fabrizio.fontana@unimi.it (F.F.); lucia.cioffi@unimi.it (L.C.); gabriela.chrostek@unimi.it (G.C.); anna.cariboni@unimi.it (A.C.); silvia.giatti@unimi.it (S.G.); 2Theoreo Srl., Via Degli Ulivi 3, 84090 Montecorvino Pugliano, Italy; abenante@theoreosrl.com (V.A.); troisi@theoreosrl.com (J.T.)

**Keywords:** intestinal macrophages, NF-κB, PPAR-α, PPAR-γ, acetate, butyrate, untargeted metabolomics, goblet cells, Claudin-5, ZO-1

## Abstract

Background: Finasteride, a 5α-reductase inhibitor commonly prescribed for androgenetic alopecia, has been linked to persistent adverse effects after discontinuation, known as post-finasteride syndrome (PFS). Symptoms include neurological, psychiatric, sexual, and gastrointestinal disturbances. Emerging evidence suggests that PFS may involve disruption of sex steroid homeostasis, neuroactive steroid deficiency (notably allopregnanolone, ALLO), and gut–brain axis alterations. Objective: This study aimed to investigate the effects of finasteride withdrawal (FW) in a rat model and evaluate the potential protective effects of ALLO on gut and hypothalamic inflammation. Methods: Adult male *Sprague Dawley* rats were treated with finasteride for 20 days, followed by one month of drug withdrawal. A subgroup received ALLO treatment during the withdrawal. Histological, molecular, and biochemical analyses were performed on the colon and hypothalamus. Gut microbiota-derived metabolites and markers of neuroinflammation and blood–brain barrier (BBB) integrity were also assessed. Results: At FW, rats exhibited significant colonic inflammation, including a 4.3-fold increase in Mφ1 levels (*p* < 0.001), a 2.31-fold decrease in butyrate concentration (*p* < 0.01), and elevated hypothalamic GFAP and Iba-1 protein expression (+360%, *p* < 0.01 and +100%, *p* < 0.01, respectively). ALLO treatment rescued these parameters in both the colon and hypothalamus but only partially restored mucosal and BBB structural integrity, as well as the NF-κB/PPARγ pathway. Conclusions: This preclinical study shows that FW causes inflammation in both the gut and hypothalamus in rats. ALLO treatment helped reduce several of these effects. These results suggest ALLO could have a protective role and have potential as a treatment for PFS patients.

## 1. Introduction

Finasteride is a 5α-reductase inhibitor, widely prescribed for androgenetic alopecia and benign prostatic hyperplasia [1]. Despite its efficacy, this drug also induces sexual, psychological, and systemic side effects during treatment as well as at withdrawal, with the occurrence of the so-called post-finasteride syndrome (PFS) [2,3,4,5,6,7].

Emerging evidence highlights the systemic disruption of steroid homeostasis as a key feature of PFS. Indeed, as reported in PFS patients and in experimental models, finasteride not only decreases the levels of 5α-reduced metabolites of testosterone, such as dihydrotestosterone, as expected, but also affects those of progesterone, like allopregnanolone (ALLO), and exerts a global effect on different steroid molecules [8,9]. These alterations, with some differences depending on the compartments observed, may occur in plasma and in cerebrospinal fluid [8], brain areas [9], and corpus cavernosum [10], as well as in the gut [11]. Steroid level alterations are related to sexual dysfunction [6,8], neuroinflammation, depression, anxiety, and alterations in gut microbiota and gut inflammation observed in PFS patients and/or experimental models [11,12,13]. In particular, since the decrease in ALLO levels observed in the gut after finasteride withdrawal (FW) correlates with an inflammatory state, as evidenced by increased levels of proinflammatory cytokines (i.e., interleukin 1-beta and tumor necrosis factor-alpha) and alterations in colonic serotonin and dopamine levels, we have recently explored whether this steroid may exert protective effects [11]. As demonstrated, ALLO administration in a rat model of PFS was able to reduce intestinal inflammation [11]. The anti-inflammatory effect of this progesterone metabolite is not surprising. Indeed, it has been demonstrated not only in the peripheral but also in the nervous system [14], where ALLO exerts a variety of neuroprotective effects in neurodegenerative and psychiatric disorders, in which neuroinflammation and systemic inflammation exert a key role [15]. In addition, in this context, it is important to highlight the existence of the gut–brain axis. This bidirectional communication network linking gastrointestinal and CNS function has emerged as a critical mediator of systemic diseases characterized by endocrine disruption, neuroinflammation, and chronic inflammation [16,17,18]. Gut dysbiosis, the disruption of the intestinal barrier, and dysregulated microbial metabolite production are critical contributors to both peripheral and central pathological processes [19]. Given the overlap between these mechanisms and the systemic effects observed in PFS, it is increasingly evident that gut dysregulation may be a fundamental contributor to the PFS pathogenesis.

Building on these concepts, the present study aims to further investigate the mechanisms of gut dysbiosis in an experimental model of PFS in adult male rats following finasteride withdrawal, as well as the potential protective effects of ALLO. Considering the strong gut–brain interplay, we will additionally assess the neuroprotective properties of ALLO in the hypothalamus, a central brain region implicated in the modulation of the gut–brain axis. Specifically, this study will examine (i) phenotypic characterization of lamina propria macrophages in the colon, including M1 subtype analysis and oxidative stress in the colon; (ii) intestinal mucosal morphology; (iii) short-chain fatty acids (SCFAs) in fecal samples as key microbiota-derived metabolites; (iv) markers of neuroinflammation and blood–brain barrier integrity in the hypothalamus; and (v) untargeted metabolomics for the identification of metabolic changes in the cecum.

## 2. Materials and Methods

### 2.1. Animals

Experiments were conducted using male adult Sprague Dawley rats (200−225 g, Charles River Laboratories, Lecco, Italy). All procedures adhered to national (D.L. No. 26, 4 March 2014) and international regulations (EEC Directive 2010/63, 22 September 2010) and were approved by the local ethics committee and the Italian Ministry of Health (authorization 261-2021-PR). Animals were housed in the Dipartimento di Scienze Farmacologiche e Biomolecolari (DiSFeB) at the Università degli Studi di Milano, Italy, and acclimated for one week. At sacrifice, rats were anesthetized with 5% isoflurane (ISO VET, La Zootecnica, Pavia, Italy), and samples were collected for analysis. Furthermore, all experimental groups underwent the same anesthetic procedure, which minimized the potential confounding effect of isoflurane on the oxidative parameters assessed in this study.

### 2.2. Treatment and Experimental Design

Twenty-four male Sprague Dawley rats were divided into three groups: control (CTR, n = 8), finasteride withdrawal (FW, n = 8), and finasteride + allopregnanolone (FW+ALLO, n = 8). Finasteride-treated rats received 3 mg/kg/day of finasteride (Sigma-Aldrich, Milan, Italy) dissolved in sesame oil and ethanol (5% *v*/*v*) for 20 days. Control rats received only the vehicle solution. After a two-week washout period, 8 finasteride-treated rats were given 3 mg/kg of allopregnanolone (ALLO, Sigma-Aldrich, Milan, Italy) every other day, while control and remaining finasteride-treated rats received only the vehicle solution, based on an experimental schedule already applied with success [11]. All treatments were administered subcutaneously. Rats were sacrificed 24 h after the last ALLO injection, 1 month after the last finasteride injection.

Macrophage isolation and analysis were performed using the lamina propria of the colon via flow cytometry; for this, intestinal tissues were harvested and incubated in Ca^2+^/Mg^2+^ free HBSS with 5% FBS and 2 mM EDTA at 37 °C for 20 min with horizontal shaking, as previously reported [20]. After removing the epithelial layer, tissues were minced and digested in HBSS/FBS with collagenase (1.5 mg/mL) and DNase (0.04 mg/mL). After incubation, the digested tissue was filtered, washed, centrifuged, and resuspended in cold PBS. Pellets were stained with Live–Dead viability/cytotoxicity stain (ThermoFisher Scientific, Milano, Italy), fixed in 4% PFA, and blocked with rat serum. Cells were stained with primary antibodies (anti-CD45, anti-CD11b, anti-CD68, and anti-CD86), washed, and analyzed by flow cytometry using the Fortessa BD X20 (BD Biosciences, Franklin Lakes, NJ, USA). For details, see Appendix A.

### 2.3. Thiobarbituric Acid Reactive Substance

The production of reactive oxygen species in the colon was measured using the thiobarbituric acid reactive substance (TBARS) assay [21]. Colon tissue (300 mg) was homogenized in 5 mL lysis buffer (Tris HCl 0.1 M, pH 7.4; EDTA 1.34 mM; glutathione 0.65 mM). The supernatant was collected, and homogenization was repeated before the final homogenization in 5 mL lysis buffer. A 100 μL aliquot of homogenate was mixed with 800 μL reaction solution (TBA in 1% phosphoric acid) and incubated at 95 °C for 45 min. The reaction was extracted with 1 mL n-butanol and centrifuged at 4000× *g* for 20 min. The supernatant was measured fluorometrically using an Enspire multiplate reader (Perkin Elmer, Waltham, MA, USA) at an excitation of 532 nm and emission of 553 nm. MDA levels were quantified using a standard curve prepared and analyzed on the same day under identical conditions.

### 2.4. RNA Extraction and Gene Expression Analysis

RNA was extracted from snap-frozen colon using the Direct-zol™ MiniPrep kit (Zymo Research, Irvine, CA, USA) following the manufacturer’s protocol, after homogenization in a Tissue Lyser (Qiagen, Milano, Italy) with EUROGOLD TriFast (Euroclone, Milano, Italy). RNA quantification was performed using a NanoDrop™ 2000 (ThermoFisher Scientific, Milano, Italy). Gene expression was assessed by TaqMan quantitative real-time PCR using a CFX96 system (Bio-Rad Laboratories, Segrate, Italy). Samples were run in duplicate using 96-well plates as multiplexed reactions, with 36B4 gene expression as an internal control (Eurofins MWG-Operon, Milan, Italy) using the Luna Universal One-Step RT-qPCR Kit (New England BioLabs Inc., Ipswich, MA, USA). Briefly, 10 μg of total RNA was combined with specific TaqMan MGB probes and primers (Eurofins MWG-Operon, Milano, Italy). Primer sequences were as follows: NF-κB forward, 5′-CTACGAGACCTTCAAGAGCATC-3′ and reverse, 5′-GATGTTGAAAAGGCATAGGGC-3′; 36B4 forward, 5′-GACTCTCGGCATGCTAACTAG-3′ and reverse, 5′-GGACATCTAAGGGCATCACAG-3′. PPAR-α (Rn00566193_m1) and PPAR-γ (Rn00440945_m1) probes were purchased from Life Technologies Italia (Milan, Italy).

### 2.5. Histological Analysis of Colon Tissues

Colon tissues were cleaned from stool in ice-cold PBS and prepared as Swiss rolls [22] and then fixed overnight in 10% neutral-buffered formalin (Bio-Optica, Milan, Italy), dehydrated, and embedded in paraffin. Sections (4 μm) were deparaffinized in xylene, rehydrated through an ethanol series (100%, 96%, 70%, 50%), and processed for histology. For H&E staining, sections were treated with Mayer’s Hemalum (H&E, Bio-Optica, 05-M06002) for 5 min and eosin–phloxine (H&E, Bio-Optica, 05-M10020) for 2 min. PAS (Bio-Optica, 04-130802) and Alcian Blue (Bio-Optica, 04-160802) staining were performed following the manufacturer’s instructions. Sections were mounted using Bio Mount HM (Bio-Optica, 05-BMHM100). Images were acquired using an Axioskop2 plus microscope (Zeiss, Oberkochen, Germany) with a THC-5.0ICE digital camera, and image analysis was conducted using ImageJ (v1.53, NIH). Crypt depth was measured in H&E-stained sections at 20× magnification, considering well-oriented crypts, with measurements taken from the crypt junction to the base of the crypt. Goblet cell quantification was performed on PAS and Alcian Blue-stained sections, counting PAS- and Alcian Blue-positive cells per crypt at 20× magnification using ImageJ, in a blinded manner. Approximately 90 crypts per rat were analyzed for crypt depth, and 60 crypts per animal for goblet cell count.

### 2.6. SCFA Quantification and Analysis

The absolute quantification of SCFAs in fecal samples was performed using LC-MS/MS, as previously described [23] (see Appendix A). Samples were extracted, derivatized, and analyzed using an API 3500 mass spectrometer (AB Sciex, Framingham, MA, USA) coupled with an HPLC system. Chromatographic separation was achieved with a Luna Omega C18 column (Phenomenex, Torrance, CA, USA), and data were processed using Analyst software (version 1.7). For details, see Appendix A.

### 2.7. Western Blotting Analysis

Hypothalami were homogenized using the Tissue Lyser (Qiagen, Milano, Italy) in lysis buffer (PBS, pH 7.4, 1% Nonidet P-40) with a protease inhibitor cocktail (Roche Diagnostics, Monza, Italy). After centrifugation at 2000× rpm for 5 min at 4 °C, the pellet was discarded, and the protein-containing supernatant was analyzed. Protein concentration was determined using the Bradford Assay (Bio-Rad, Segrate, Italy), and equal concentration aliquots were prepared and heated to 100 °C for 5 min. Samples were separated on Criterion acrylamide gels (Bio-Rad, Segrate, Italy) and transferred to nitrocellulose membrane. Membranes were blocked with 10% non-fat dry milk or 5% bovine serum albumin (BSA) for 1 h at room temperature and incubated with primary antibodies (GFAP, Invitrogen (Thermo Fisher Scientific, Waltham, MA, USA), #13-0300; Iba-1, Proteintech Germany GmbH, Martinsried, Germany , #26177-1-AP; ZO-1, Invitrogen, #61-7300; Claudin-5, Invitrogen, #34-1600; GAPDH, SigmaAldrich #G9545, diluted 1:1000 in PBS-Tween 0.1% (PBST) with 5% non-fat dry milk or 5% BSA. After washing, membranes were incubated with horseradish peroxidase-conjugated secondary antibodies (anti-mouse or anti-rabbit, depending on the primary antibody) diluted in PBST supplemented with either 2.5% BSA or 1% non-fat milk. Protein detection was performed using the enhanced chemiluminescence (ECL) method (Bio-Rad, Segrate, Italy), acquired with the ChemiDoc™ XRS+ system (Bio-Rad, Segrate, Italy), and analyzed with Image Lab™ software (v. 5.2.1, Bio-Rad, Segrate, Italy). Protein levels were normalized to GAPDH and expressed as percentages relative to the mean value of the control group, which was set to 100%.

### 2.8. Metabolomic Profiling and Analysis

Untargeted GC-MS-based metabolomic profiling was performed on cecal samples using the MetaboPrep GC kit (Theoreo srl, Salerno, Italy), following protocols previously described by Troisi et al. (see Appendix A). After extraction, purification, and derivatization, metabolites were separated via gas chromatography and detected using mass spectrometry. Identification was based on spectral matching against the NIST-2014 library, with confidence levels assigned per MSI guidelines. A total of 248 metabolites were annotated, with further analysis restricted to those with VIP > 1.5, confirmed via external standards. Preprocessed data were analyzed using MetaboPredict^®^ software (version 1.2.1). Normalization procedures accounted for internal standard response and sample weight. Class separation was assessed using PLS-DA with permutation testing, and relevant metabolites were prioritized using VIP scores. Metabolic pathway analysis was conducted using MetPA with enrichment and topology-based approaches, referencing KEGG pathways. For details, see Appendix A.

### 2.9. Statistics

Data were analyzed by one-way ANOVA to compare differences between the three experimental groups: control, FINA, and FINA + ALLO. The effect of the treatment with ALLO was analyzed using one-way ANOVA followed by uncorrected Fisher’s LSD post hoc test. Analyses were performed using Prism, version 7.0a (GraphPad Software Inc., San Diego, CA, USA), after checking for normal distribution with the Kolmogorov–Smirnov test. *p* < 0.05 was considered significant.

## 3. Results

### 3.1. Allopregnanolone Treatment Reverses Finasteride Withdrawal-Induced Macrophage (CD86^+^) Polarization

Macrophages (Mφ) were isolated from the lamina propria of the colon via mechanical dissociation, followed by flow cytometry analysis. Cells were identified using the following markers: CD45 for pan-leukocytes, CD11b for monocytes/macrophages, CD68 for macrophages, and CD86 for Mφ1. After gating out dead cells, the macrophage population was selected based on CD11b and CD68 expression, while CD86 was used to assess the activation status of macrophages.

Notably, FW significantly increased macrophage infiltration in the colon compared to controls (Figure 1A). A downward trend was observed with ALLO treatment, suggesting a potential modulatory effect.

As shown in Figure 1B, a significant increase in the proportion of CD86-expressing macrophages was observed in FW animals, indicating a shift toward a proinflammatory Mφ1 phenotype. Interestingly, ALLO treatment significantly reduced CD86 expression compared to FW animals, further supporting a modulatory effect on the inflammatory response (Figure 1B).

### 3.2. Finasteride Withdrawal Increases Oxidative Stress and Decreases PPAR Gene Expression in the Colon—Activation of Protective Mechanisms by Allopregnanolone

To assess oxidative stress in the colon, we measured TBARS levels, a marker of lipid peroxidation. FW rats showed a significant increase in TBARS, indicating elevated oxidative stress compared to controls (Figure 2A). Treatment with ALLO led to a reduction in TBARS levels, though this decrease did not reach statistical significance (*p* = 0.056), indicating a potential trend toward protective effects against oxidative damage. Gene expression analysis of NF-κB, PPAR-α, and PPAR-γ was performed to explore the molecular mechanisms underlying the effects of FW and ALLO treatment (Figure 2B). NF-κB expression was significantly higher in FW rats, indicating increased inflammation, while ALLO did not affect this parameter. PPAR-α expression was lower in FW rats but significantly increased with ALLO treatment, suggesting a protective effect. In contrast, PPAR-γ expression was not modulated by ALLO, despite being affected by FW.

### 3.3. Finasteride Withdrawal Alters the Intestinal Epithelium Morphology Affecting Mucin Composition and Crypt Depth—Protective Effects of Allopregnanolone

Histological analysis was conducted to assess goblet cell distribution and crypt depth in the colon (Figure 3). Alcian Blue (AB) staining, which highlights acidic mucins, showed a decrease in goblet cell content in the FW group, indicating impaired intestinal function (Figure 3A). In contrast, PAS staining revealed an increase in neutral mucins in the FW group (Figure 3B). ALLO treatment significantly restored neutral mucin levels to baseline but did not significantly affect acidic mucins. Furthermore, H&E staining revealed increased crypt depth in FW rats, which was significantly reversed by ALLO treatment, restoring crypt depth to baseline levels (Figure 3C).

### 3.4. Finasteride Withdrawal Decreases Acetate and Butyrate Fecal Levels and Induces Hypothalamic Neuroinflammation—Protective Effects of Allopregnanolone Treatment

Fecal SCFA analysis (Figure 4A) showed a significant reduction in butyrate and acetate in the FW group compared to controls. ALLO treatment significantly restored both butyrate and acetate levels. Other SCFAs did not show significant changes.

To evaluate neuroinflammation and blood–brain barrier (BBB) integrity in the hypothalamus, we performed Western blotting analysis for Iba-1 (microglial marker), GFAP (astrocyte marker), and tight junction proteins (ZO-1 and Claudin-5). Western blotting in the hypothalamus revealed increased Iba-1 and GFAP expression in the FW group, indicating neuroinflammation. ALLO treatment significantly reduced these markers. ZO-1 expression was increased, while Claudin-5 was decreased in the FW group (Figure 4B). ALLO partially restored ZO-1 but did not significantly restore Claudin-5 levels. Representative blots are shown in Figure 4C.

### 3.5. Metabolic Shifts in the Cecum Induced by Finasteride Withdrawal

To further elucidate the metabolic alterations associated with FW, an untargeted metabolomics analysis was conducted on cecal content samples, providing a comprehensive overview of host–microbiota interactions and gut microbial metabolic activity. The analysis revealed a markedly distinct metabolomic profile in the FW group compared to the controls, underscoring significant shifts in gut biochemistry induced by the FW.

Among the metabolites modulated (selected as the ones with a VIP-score ≥ 1.5), between the two groups were various amino acids and related compounds (e.g., serine, aspartic acid, and aminomalonic acid), monosaccharides and sugar alcohols (glucose, arabinose, N-acetyl-D-glucosamine, and lactulose), lipid- and fatty acid-related metabolites (glycerol monostearate, valeric acid, and glyceric acid), organic acids (malic acid and tartaric acid), nucleoside thymidine, and other low-molecular-weight compounds such as diethylene glycol and 2-methylbenzoic acid.

Notably, three metabolites—pentanedioic acid (glutaric acid), allocholic acid, and pyroglutamic acid—were found to be increased in the FW group relative to controls. In contrast, all other metabolites exhibited a general decrease in abundance following FW, indicating a widespread downregulation of various metabolic pathways.

This trend was clearly illustrated in the PLS-DA 3D score plot (Figure 5A), which demonstrated a distinct separation between the CTR and FW groups, reflecting FW-induced metabolic divergence. The VIP scores (Figure 5C) identified key metabolites contributing to this separation; furthermore, the heatmap visualization (Figure 5B) reinforced these findings, with shades of white representing decreased metabolite abundance and black indicating increased levels.

Together, these findings highlight the substantial metabolic remodeling induced by FW and point to specific biochemical pathways—particularly those involving fatty acid metabolism, bile acid derivatives, and glutamate cycling—that may be differentially regulated in response to the finasteride suspension.

## 4. Discussion

A decline in the levels of neuroactive steroids (i.e., steroids synthesized in the nervous system as well as in peripheral endocrine glands and affecting nervous functions) has been widely proposed as a contributing factor to the development of mood disorders and sexual dysfunction [24,25]. In particular, low levels of ALLO (i.e., a metabolite of progesterone) in peripheral blood or cerebrospinal fluid have been associated with major depressive disorder (MDD) [26], post-traumatic stress disorder (PTSD) [27], premenstrual dysphoric disorder (PDD) [28], postpartum depression (PPD) [29], and female [30] and male sexual dysfunction, including PFS [8]. These molecules have attracted increasing attention for their potential therapeutic role in depressive disorders, exhibiting properties that distinguish them from conventional antidepressants [31]. This perspective has been further supported by the FDA’s recent approval of an oral ALLO-based drug for the treatment of PPD [32].

Recent studies have shown that human gut bacteria can convert biliary corticoids into progestins, including ALLO, which may influence peripheral ALLO levels and, consequently, modulate brain function and behavior [33]. While ALLO’s effects have mainly been studied in CNS pathophysiology, including clinical studies in PFS [8] and in its experimental model [9], the literature data indicate that the GABA_A_ receptor is also expressed in the enteric nervous system and peripheral macrophage surfaces, where it may play a role in modulating gut motility and inflammation [34,35,36,37]. ALLO primarily binds to GABA_A_ receptors, its main target, and this builds on our previous findings that ALLO treatment during FW restored GABA_A_ receptor subunit alterations and anti-inflammatory cytokine levels in the colon [11]. Here, we further explored its protective effects.

In this PFS rat model, we showed that FW significantly increased macrophage numbers in the colon compared to controls. Notably, about 50% of these macrophages were polarized to the proinflammatory M1 phenotype. This polarization, along with inflammation in the lamina propria, was linked to elevated oxidative stress and NF-kB levels, suggesting that FW may drive inflammatory and oxidative damage in the colon. These changes are commonly associated with inflammatory bowel diseases (IBDs) [38] and may be potentially linked to autoimmune diseases and cancer [39,40]. Eight ALLO treatments, given every other day during FW, significantly reduced M1 macrophages to 17% of the total, indicating a direct role of ALLO in modulating the immune response in the colon. Given that our previous studies showed ALLO accumulation in the colon with this treatment schedule [11], it is likely that macrophage polarization is directly influenced by ALLO, though alternative mechanisms cannot be ruled out. FW increased colonic NF-kB expression and decreased PPAR-α and PPAR-γ levels, mirroring inflammatory signaling observed in IBDs [41,42]. Notably, ALLO treatment specifically restored PPAR-α expression. This is significant because PPAR-α has been implicated in stimulating neuroactive steroid biosynthesis in the brain [43,44,45], and its upregulation may represent a key mechanistic pathway through which ALLO exerts anti-inflammatory effects. In line with a previous hypothesis [43], PPAR-α activation may potentially counteract NF-κB signaling, suggesting that ALLO’s restorative action on PPAR-α could underlie its ability to suppress colonic inflammation. To date, no studies have explored whether ALLO modulates PPAR expression in the colon, making these findings particularly novel. Together, these results point to a potential PPAR-α-mediated mechanism by which ALLO mitigates FW-induced inflammatory signaling in this PFS model. Additionally, we observed remarkable changes in goblet cell populations, characterized by a reduction in acid-secreting goblet cells and a concomitant increase in neutral-secreting goblet cells. Notably, acid goblet cells primarily contribute to mucin production and host homeostasis [46], and their reduction suggests a potential impairment in mucus barrier integrity, which may exacerbate epithelial vulnerability to inflammation [47], as reported in IBDs [48] and also in neurodegenerative disorders [49]. On the other hand, the increase in neutral goblet cells could represent a compensatory response, although its functional implications remain unclear. Mucus is essential for gut homeostasis, aiding food passage, protecting the epithelium, and serving as a barrier against pathogens, toxins, and irritants. Mucus viscosity plays a key role; less viscous mucus from neutral mucins spreads easily, reducing protection, while more viscous mucus forms a stable barrier [50]. The observed shift from acidic to neutral mucins may impair the protective function of the mucus layer, as acidic mucins are more effective in forming a viscous and negatively charged barrier that limits microbial adhesion and penetration. A relative increase in neutral mucins could therefore alter microbial colonization patterns and compromise barrier integrity, potentially promoting inflammation. These findings were further supported by structural changes in colonic crypt depth, including elongation, which are typical indicators of intestinal inflammation in rodents [51]. Interestingly, ALLO treatment reversed the FW-induced changes in crypt length and neutral mucins but not in acidic mucins, probably because it only partially restored morphological integrity without fully correcting mucus composition.

Despite the complex PFS patient phenotype still not being well characterized in the animal model, adult PFS rats exhibit, like patients [8,13], a depressive-like behavior associated with gut dysbiosis [12]. Consistent with this, we observed decreased butyrate and acetate levels in fecal samples, which are commonly linked to anxiety and depression in both rodents and patients [52,53,54,55]. In PFS patients, reduced abundance of *Faecalibacterium* spp., a butyrate-producing bacterium [56], is believed to play a key role in the gut–brain axis. However, research on the effects of chronic low-level butyrate and other SCFA exposures is limited [57]. Nonetheless, evidence suggests that butyrate production by the gut microbiota influences peripheral immune function, shaping the brain’s immune environment [58]. Butyrate and acetate have been demonstrated to exert anti-inflammatory effects on microglia [59,60,61,62], aligning with our findings that ALLO treatment restores Iba-1 and GFAP protein expression in the hypothalamus altered by FW. Therefore, this set of analyses suggests a potential role of ALLO in modulating inflammation and neuroinflammation in this PFS animal model. Given that butyrate influences both peripheral and central immune responses, future studies targeting these pathways could reveal therapeutic potential for ALLO in neuroinflammation. Specifically, vagotomy studies could clarify whether ALLO’s effects on butyrate levels and glial activation are mediated through gut–brain communication or alternative mechanisms. Our previous preclinical study suggested that finasteride induced altered tight junctions (TJs) in the hypothalamus [63]. TJs regulate the permeability of biological barriers, including the BBB, and are essential for neural function [64]. Moreover, alterations in TJs, particularly Claudin-5 and ZO-1, may contribute to neurobiological dysfunctions [65,66]. In this PFS model, low Claudin-5 and high ZO-1 expression could suggest a disorganization or alteration in TJ formation. A reduction in Claudin-5 may compromise TJ integrity, while high ZO-1 expression might reflect a compensatory response or a dysfunction in the TJs. These changes could increase permeability compared to the more organized TJs seen in the median eminence tanycytes [67]. The alterations in TJs could also be a consequence of HPA dysfunction, which can be modulated by negative feedback from ALLO, as reported in MDD and PTSD [68]. Future studies, such as those using Evans blue dye or FITC–dextran injections, are needed to confirm BBB disruption and explore how ALLO restores TJ levels in the hypothalamus. Our results also suggest that, in this PFS experimental model, microbiota-derived metabolites, beyond SCFAs, may modulate the collateral effects induced by FW. Our untargeted metabolomic analysis revealed significant alterations, particularly in serine and glutaric acid levels. Serine is crucial for regulating cytokine production and macrophage polarization [69], and its deficiency exacerbates inflammation and oxidative stress through the gut–microbiota–brain axis [70]. Moreover, changes in glutaric acid levels in the colon are consistent with findings that glutaric acid disrupts neurotransmission, redox homeostasis, bioenergetics, BBB integrity, brain vasculature, and myelination while also inducing reactive astrogliosis and neuronal death [71]. Future studies should focus on the molecular mechanisms underlying these FW–microbiota interactions.

The limitations of these findings stem from the preclinical nature of the study, which precludes direct clinical correlation, and the incomplete characterization of the PFS animal model. While the study highlights novel insights into the protective actions of ALLO, alternative mechanisms underlying macrophage polarization and the effects of this molecule on inflammation cannot be excluded. Furthermore, a deeper understanding is required of how ALLO influences PPAR-α expression, its role in neuroactive steroid biosynthesis, and its capacity to restore barrier integrity. Lastly, the etiopathogenesis of the observed alterations warrants further investigation. For instance, the impact of microbiota-derived metabolites, such as butyrate and acetate, on neuroinflammation and glial activation remains unclear, as does the potential compensatory role and functional significance of neutral goblet cell responses.

## 5. Conclusions

The aim of this study was to evaluate the therapeutic potential of ALLO treatment in modulating intestinal inflammation, immune response, and gut–brain axis function in the context of PFS. Our results demonstrate that ALLO has a beneficial effect on markers of hypothalamic neuroinflammation, suggesting a positive action on the CNS. However, although no significant improvements were observed in Claudin-5 protein levels, an improvement in ZO-1 was noted, indicating that ALLO may influence specific aspects of the BBB structure without modulating all components of its integrity. Additionally, ALLO treatment positively influenced parameters such as macrophage polarization (M1) and PPARα expression, with beneficial effects primarily observed in the modulation of inflammation and improvement in mucosal integrity. However, not all parameters related to mucosal morphology showed significant improvement, suggesting that ALLO’s anti-inflammatory effect is not pleiotropic. In particular, while the FW modulates NF-κB and PPARα/γ pathways, ALLO treatment appears to effectively restore only PPARα, without significantly affecting NF-κB/PPARγ. These results suggest that ALLO’s therapeutic effect may be primarily mediated by PPARα, as well as by metabolites like butyrate and acetate, rather than by a widespread effect on multiple molecular pathways.

Future studies should focus on a deeper understanding of the molecular mechanisms underlying the protection observed in our PFS model, as well as the underlying causes of intestinal inflammation that develop following drug withdrawal. Moreover, a potential link between the alteration reported here and its possible behavioral consequences will be investigated in future studies. These insights could help identify more specific therapeutic targets and optimize treatment strategies for patients with PFS.

## Figures and Tables

**Figure 1 biomolecules-15-01044-f001:**
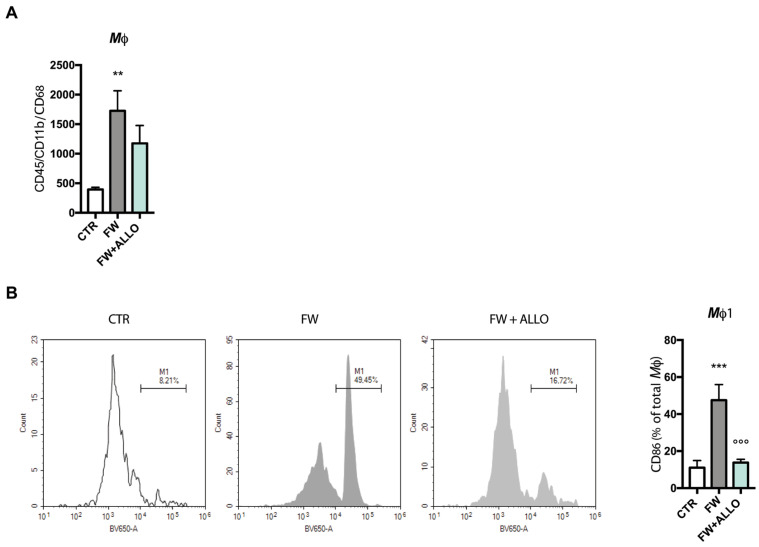
Isolation and flow cytometry analysis of CD45^+^/CD11b^+^/CD68^+^ cells from the lamina propria, including quantification of Mφ1 (CD86^+^) cell polarization: (**A**) histogram shows the total number of macrophages isolated from the lamina propria of the colon in control (CTR), finasteride withdrawal (FW), and ALLO-treated groups (FW + ALLO). F (2, 12) = 13.77, *p* = 0.0008; (**B**) representative flow cytometry graphs showing the expression of CD86^+^ cells across experimental groups. F (2, 12) = 6.40, *p* = 0.01. The columns represent the mean ± SEM (n = 5 for each group). The effect of ALLO was analyzed using one-way ANOVA followed by uncorrected Fisher’s LSD post hoc test; statistical significance: ** *p* < 0.01, *** *p* < 0.001 vs. control group; °°° *p* < 0.001 vs. FW group.

**Figure 2 biomolecules-15-01044-f002:**
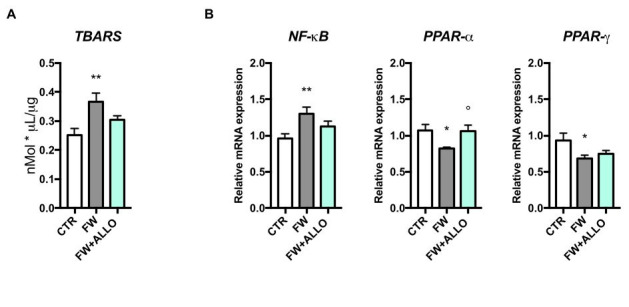
Measurement of TBARS levels and expression of inflammatory markers in colon samples: (**A**) levels of TBARS as a marker of lipid peroxidation, in the colon of control (CTR), finasteride withdrawal (FW), and ALLO-treated groups (FW + ALLO). F (2, 18) = 6.846, *p* = 0.0061; (**B**) the columns represent the mean ± SEM after normalization with 36B4 mRNA. NF-κB (F (2, 18) = 4.749, *p* = 0.0221), PPAR-α (F (2, 18) = 4.165, *p* = 0.0326), and PPAR-γ (F (2, 18) = 3.566, *p* = 0.0496). The effect of ALLO was analyzed using one-way ANOVA followed by uncorrected Fisher’s LSD post hoc test; statistical significance: * *p* < 0.05, ** *p* < 0.01 vs. control group; ° *p* < 0.05 vs. FW group, n = 7.

**Figure 3 biomolecules-15-01044-f003:**
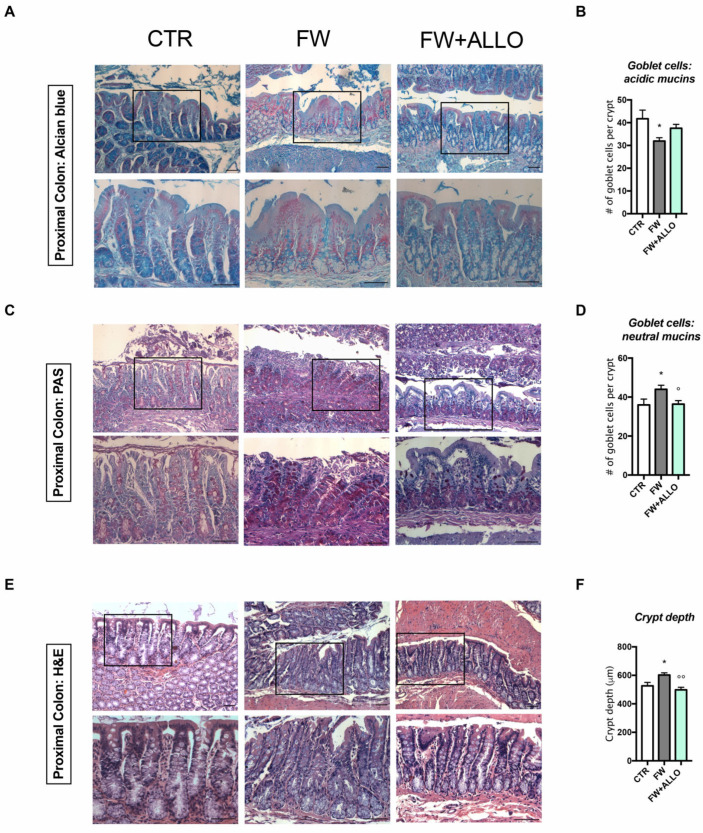
Histological staining of colon tissue with AB, PAS, and H&E, and quantification of crypt depth and mucin composition. Representative histological examination of colon sections to assess tissue architecture and cellular morphology: (**A**,**C**,**E**) representative AB, PAS, and H&E staining of the colon sections. Black boxes indicate areas shown at higher magnification below each corresponding panel. Scale bar: 250 μm (low magnification), 125 μm (high magnification); (**B**) histogram representing the quantification of goblet cells (acidic mucins) across different treatment groups, F (2, 15) = 3.752, *p* = 0.0477; (**D**) histogram representing the quantification of goblet cells (neutral mucins) across different treatment groups, F (2, 15) = 3.708, *p* = 0.0492; (**F**) histogram provides the measure of crypt depth across different treatment groups, F (2, 14) = 8.565, *p* = 0.0037. Error bars represent mean ± SEM. The effect of ALLO was analyzed using one-way ANOVA followed by uncorrected Fisher’s LSD post hoc test; statistical significance: * *p* < 0.05 vs. control group; ° *p* < 0.05, °° *p* < 0.01 vs. FW group (n = 5–6).

**Figure 4 biomolecules-15-01044-f004:**
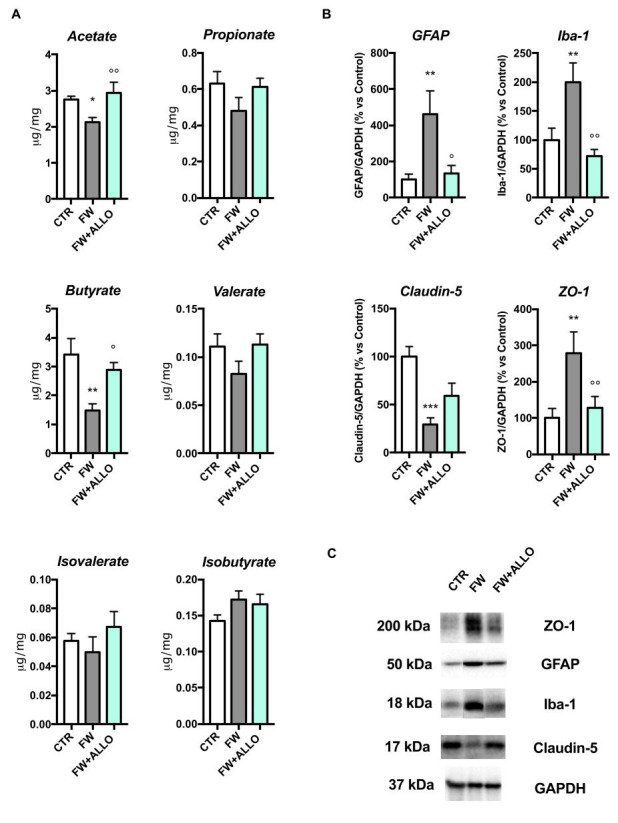
SCFA analysis in fecal samples by LC-MS/MS and neuroinflammation and BBB integrity markers by Western blotting in the hypothalamus: (**A**) quantification of acetate: F (2, 18) = 4.731, *p* = 0.0223; propionate: F (2, 18) = 1.705, *p* = 0.2099; butyrate: F (2, 18) = 7.011, *p* = 0.0056; valerate: F (2, 18) = 1.895, *p* = 0.1791; isovalerate: F (2, 18) = 0.8975, *p* = 0.4250; isobutyrate: F (2, 18) = 1.684, *p* = 0.2136 in fecal samples from control (CTR), finasteride withdrawal (FW), and ALLO-treated (FW + ALLO) groups, analyzed by LC-MS/MS; (**B**) Western blot analysis of hypothalamic neuroinflammatory markers (GFAP: F (2, 18) = 6.237, *p* = 0.0088, Iba-1: F (2, 18) = 8.141, *p* = 0.0030) and BBB integrity markers (Claudin-5: F (2, 18) = 11.33, *p* = 0.0007, ZO-1: F (2, 18) = 5.46, *p* = 0.0140). Representative blots are shown in (**C**). In the densitometric analysis, protein expression levels are normalized to housekeeping protein (i.e., GAPDH). Data are expressed as mean ± SEM, n = 7 for each group. * *p* < 0.05, ** *p* < 0.01, *** *p* < 0.001 vs. control group; ° *p* < 0.05, °° *p* < 0.01 vs. FW group.

**Figure 5 biomolecules-15-01044-f005:**
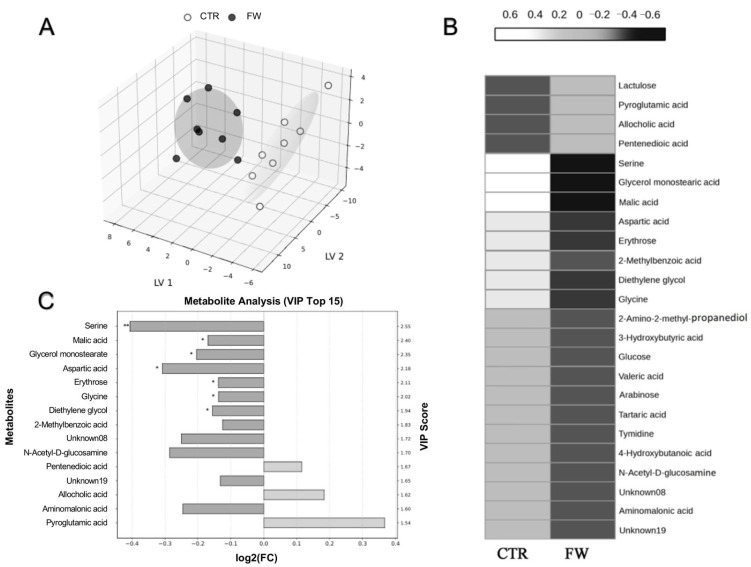
Untargeted metabolomic analysis of the cecum by LC-MS/MS: (**A**) PLS-DA 3D plot; (**B**) heatmap; (**C**) VIP scores showing notable differences between CTR and FW groups (n = 8). * *p* < 0.05; ** *p* < 0.01.

## Data Availability

The data supporting the reported results are available upon request from the corresponding author.

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
