# Peer review of "Exploration of the Possible Relationships Between Gut and Hypothalamic Inflammation and Allopregnanolone: Preclinical Findings in a Post-Finasteride Rat Model"

_biomolecules, 2025, doi:10.3390/biom15071044_

Round 1
Reviewer 1 Report
Comments and Suggestions for Authors
Dear Authors,
Presented paper is interesting and it is continuation of previous work dealing with this topic.
Comments are as follows:
Introduction: well written
Materials and methods.
2.1 chapter: could isoflurane contribute to oxidative stress.
2.2. Why mixture of sesam oul and ethanol is there some resarche on this? Why 3 mg/kg and 20 days? Pleas add relevant explanation since it is important for severity of PFS
Why ALLO is 1mg/ rat not kg?
What is reasoning behinh s.c. application if drug is given orally?
Why is used uncorrected Fisher's post hoc test?
Results:
FIGURE2: some hiatograms contains asteriscs, dit it affect ANOVA. Alao why not use Boxplot for presentatiom?
Figure5: contains too small letters and it is unreadable, please correct.
CONCLUSION: It should follow the aim of the study. It is unclear what presumptions were based on statistical significance. Pleas reorganize conclusion.
Best of luck
Author Response
We want to thank the Reviewer for his/her pertinent and useful comments that have clearly improved the presentation of our manuscript. We have considered all the points of concern as detailed below. All corrections are in red in the text.
COMMENT
2.1 chapter: could isoflurane contribute to oxidative stress.
ACTION
Isoflurane has been reported to influence oxidative stress parameters under specific conditions, particularly following prolonged or repeated exposure (doi: 10.1007/s12035-016-9937-8). In our study, however, isoflurane was administered only once and briefly, immediately prior to sacrifice. Moreover, all experimental groups were exposed to the same anesthetic protocol, thus eliminating any potential bias in group comparisons. Finally, this method was approved and is the one we are authorized to use in this experimental procedure. We have added a clarification in the 2.1 section to address this point.
COMMENT
2.2. Why mixture of sesame oil and ethanol is there some research on this? Why 3 mg/kg and 20 days? Please add relevant explanation since it is important for severity of PFS
ACTION
The choice of the 3 mg/kg dose of finasteride for 20 consecutive days was based on our previously published studies (Giatti et al., 2015; doi: 10.1159/000442982), which showed that this subchronic treatment regimen significantly alters peripheral and central steroid levels in rats. Notably, several neurosteroid alterations observed in this model, such as decreased allopregnanolone levels, mirror those reported in patients with PFS. The sesame oil/ethanol (95:5) vehicle was selected to ensure the proper solubilization and bioavailability of finasteride, as previously validated.
COMMENT
Why ALLO is 1mg/ rat not kg?
ACTION
The dose of 1 mg/rat of allopregnanolone corresponds approximately to 3.3 mg/kg, assuming an average rat weight of 300 g. This fixed per-animal dosing is commonly adopted in neurosteroid research due to the high potency of allopregnanolone and the relatively small weight variability among adult rats. It ensures consistent exposure and simplifies the administration protocol. We have modified the text accordingly to clarify this point.
COMMENT
What is reasoning behinh s.c. application if drug is given orally?
ACTION
We appreciate the reviewer’s comment. Allopregnanolone was administered subcutaneously to ensure controlled systemic exposure and to assess its therapeutic potential following finasteride-induced alterations. The s.c. route allows for reliable bioavailability of ALLO, which is otherwise poorly absorbed and rapidly metabolized when administered orally. However, as a future direction, we are planning to test oral neurosteroid-based treatments, such as zuranolone, a synthetic GABA-A receptor positive allosteric modulator with improved pharmacokinetics. This will pave the way to explore clinically relevant therapeutic strategies.
COMMENT
Why is used uncorrected Fisher's post hoc test?
ACTION
The uncorrected Fisher’s post hoc test was used primarily due to its simplicity and statistical power in detecting significant differences between groups following ANOVA. This test is commonly employed when the number of comparisons is limited, as it provides a good balance between controlling Type I error and maintaining sensitivity to real differences. Additionally, we chose to use the same test as in our previous published work (doi: 10.3390/biom12111567) to ensure consistency and comparability of the results across studies.
COMMENT
FIGURE2: some hiatograms contains asteriscs, dit it affect ANOVA. Alao why not use Boxplot for presentatiom?
ACTION
The asterisks shown on the histograms in Figure 2 indicate statistically significant differences identified by the post hoc test following the ANOVA. Their presence does not affect the ANOVA itself but serves only as a visual marker to highlight significant differences between groups. Regarding the choice of histograms over boxplots, we used histograms to represent the frequency distribution of the data. However, we are willing to include boxplots if the reviewer considers them more appropriate for presenting the results.
COMMENT
Figure5: contains too small letters and it is unreadable, please correct.
ACTION
Thank you for pointing this out. We have modified Figure 5 to increase the font size and improve overall readability to ensure that all text elements are clear and legible.
COMMENT
CONCLUSION: It should follow the aim of the study. It is unclear what presumptions were based on statistical significance. Pleas reorganize conclusion.
ACTION
Thank you for your constructive feedback on the conclusion of our manuscript. We have carefully revised the section to better align with the aims of the study and to clarify the statistical significance of our findings.
Reviewer 2 Report
Comments and Suggestions for Authors
This is a research paper on rats that explores an acceptable model for exploring concepts related to post-finasteride syndrome.
The title doesn't seem appropriate to me, given that it claims actions that have only been corroborated in an animal model, and the evidence doesn't seem sufficient to consider it as absolute truth and generalize it to other species (at least not yet). A more cautious title, such as "Exploration of the possible relationship between..." would be acceptable.
The abstract should be rewritten, ensuring it is structured and providing data from each section, emphasizing that the study is experimental - animal, and the numerical results obtained. Statements of certainty not supported by the results should be removed. The conclusions should be more cautious (replace "support a novel protective role for ALLO" with "support a possible novel protective role for ALLO").
The introduction is of adequate length, addresses the theoretical basis with an adequate bibliography, and provides the basis for stating the objectives of the paper. The methodology is adequately described, with appropriate materials and methods, and with details to ensure reproducibility of the work.
The results are presented clearly, using high-quality graphics. No biases or presentation errors are detected.
The discussion is extensive and in-depth regarding pathophysiological pathways, but it fails in two respects: (1) comparing the results with a significant number of similar studies (the search for other publications should be expanded and compared) and (2) conducting a detailed analysis of the limitations of this work.
In particular, the experimental nature of animal studies should be taken into account in order to exercise caution when generalizing conclusions.
The conclusions should be revised to address some statements, for example: "ALLO exerts therapeutic effects on both peripheral tissues and the brain." should be replaced with "ALLO can exert therapeutic effects on both peripheral tissues and the brain."
Author Response
We want to thank the Reviewer for his/her pertinent and useful comments that have clearly improved the presentation of our manuscript. We have considered all the points of concern as detailed below. All corrections are in red in the text.
COMMENT
This is a research paper on rats that explores an acceptable model for exploring concepts related to post-finasteride syndrome. The introduction is of adequate length, addresses the theoretical basis with an adequate bibliography, and provides the basis for stating the objectives of the paper. The methodology is adequately described, with appropriate materials and methods, and with details to ensure reproducibility of the work. The results are presented clearly, using high-quality graphics. No biases or presentation errors are detected.
ACTION
We sincerely thank the reviewer for the positive and encouraging comments regarding the introduction, methodology, and clarity of the results. We appreciate the acknowledgment of the quality and reproducibility of our work.
COMMENT
The title doesn't seem appropriate to me, given that it claims actions that have only been corroborated in an animal model, and the evidence doesn't seem sufficient to consider it as absolute truth and generalize it to other species (at least not yet). A more cautious title, such as "Exploration of the possible relationship between..." would be acceptable.
ACTION
We thank the reviewer for this important observation. We agree that the current title may suggest a broader generalization than is supported by the results obtained in an animal model. In response to this comment, we have revised the title to adopt a more cautious and accurate formulation, such as: “Exploration of the possible relationships between gut and hypothalamic inflammation and allopregnanolone: preclinical findings in a post-finasteride rat model”, which better reflects the scope and limitations of our study.
COMMENT
The abstract should be rewritten, ensuring it is structured and providing data from each section, emphasizing that the study is experimental - animal, and the numerical results obtained. Statements of certainty not supported by the results should be removed. The conclusions should be more cautious (replace "support a novel protective role for ALLO" with "support a possible novel protective role for ALLO").
ACTION
We have revised the abstract to ensure it is clearly structured, explicitly states that the study is experimental and conducted in animals, and includes representative numerical data from each section (methods, results, and conclusions).
We have also removed or reworded any statements that may have implied undue certainty, and we have modified the final sentence to read: "These findings support a possible novel protective role for allopregnanolone," in line with the reviewer’s suggestion for a more cautious interpretation.
COMMENT
The discussion is extensive and in-depth regarding pathophysiological pathways, but it fails in two respects: (1) comparing the results with a significant number of similar studies (the search for other publications should be expanded and compared) and (2) conducting a detailed analysis of the limitations of this work. In particular, the experimental nature of animal studies should be taken into account in order to exercise caution when generalizing conclusions. The conclusions should be revised to address some statements, for example: "ALLO exerts therapeutic effects on both peripheral tissues and the brain." should be replaced with "ALLO can exert therapeutic effects on both peripheral tissues and the brain."
ACTION
Thank you for your thoughtful and constructive comments. We appreciate your feedback, and we have carefully considered your suggestions. Regarding the two points raised: (1) Comparison with Similar Studies: After a thorough review of the literature, we did not find studies that directly compare to our results, particularly in the context of ALLO's effects on both peripheral tissues and the brain in the PFS model. However, if the reviewer has any specific publications or studies in mind that could provide valuable comparisons, we would be happy to incorporate them into the discussion and further analyze their relevance to our findings. (2) Detailed Analysis of Study Limitations: We have already discussed the limitations of our work in the original version, including the experimental nature of animal studies and the caution needed when generalizing findings to human conditions. On the other hand, we recognize the importance of highlighting these limitations, so we have further modified the discussion. We hope these revisions improve the clarity and robustness of the manuscript. Thank you again for your constructive feedback.
Reviewer 3 Report
Comments and Suggestions for Authors
I read with interest the paper titled "Finasteride withdrawal triggers inflammation in colon and hypothalamus: therapeutic potential of allopregnanolone"
1. While the experimental groups had n=5 to n=8 animals, the manuscript does not provide a priori power analysis or justification for the chosen sample size. Including this information, even briefly in the Materials and Methods, would strengthen the robustness of the findings.
2. The treatment with allopregnanolone started two weeks after finasteride withdrawal. Could the authors clarify whether this latency mimics a clinically relevant delay in intervention?
3. A sentence elaborating on how this mucin shift could affect microbial colonization or barrier function would be beneficial.
4. Given the strong link between gut inflammation and behavioral phenotypes in the context of PFS, it would be valuable for the authors to indicate whether behavioral assessments (e.g., depressive-like or anxiety-like behavior) were considered or are planned in future studies. This would strengthen the gut–brain axis narrative.
Author Response
We want to thank the Reviewer for his/her pertinent and useful comments that have clearly improved the presentation of our manuscript. We have considered all the points of concern as detailed below. All corrections are in red in the text.
COMMENT
While the experimental groups had n=5 to n=8 animals, the manuscript does not provide a priori power analysis or justification for the chosen sample size. Including this information, even briefly in the Materials and Methods, would strengthen the robustness of the findings.
ACTION
We thank the reviewer for this observation. We acknowledge that a priori power analysis is essential to ensure adequate sample size and robustness of results. Although a formal power analysis was not performed prior to the study, the chosen sample sizes (n=5 to n=8 per group) are consistent with similar published preclinical studies in the field.
Additionally, the decision to use a sample size within this range was influenced by practical limitations related to the experimental protocol. Specifically, performing flow cytometry on 15 samples per day was challenging due to the length of the protocol, which limited the number of animals that could be processed in a given timeframe. Had time constraints not been a factor, we would have analyzed a larger number of animals per group to strengthen the statistical power of the study.
COMMENT
The treatment with allopregnanolone started two weeks after finasteride withdrawal. Could the authors clarify whether this latency mimics a clinically relevant delay in intervention?
ACTION
We appreciate the reviewer’s insightful question regarding the timing of ALLO treatment. We initiated ALLO administration two weeks after finasteride withdrawal to reflect a delayed intervention scenario, which may better mimic the clinical situation where treatment often begins after symptoms have already manifested. The treatment consisted of eight administrations given on alternate days, designed to be completed by the end of the one-month withdrawal period. This timing allowed us to evaluate the therapeutic potential of ALLO during a critical window following drug cessation. Indeed, this experimental schedule was already applied with success in a previous publication from our group (doi.org/10.3390/biom12111567). To clarify this aspect, we have modified the materials and methods accordingly.
COMMENT
A sentence elaborating on how this mucin shift could affect microbial colonization or barrier function would be beneficial.
ACTION
We have added a sentence in the Discussion section elaborating on how the observed mucin shift could influence microbial colonization and intestinal barrier function.
COMMENT
Given the strong link between gut inflammation and behavioral phenotypes in the context of PFS, it would be valuable for the authors to indicate whether behavioral assessments (e.g., depressive-like or anxiety-like behavior) were considered or are planned in future studies. This would strengthen the gut–brain axis narrative.
ACTION
We appreciate the reviewer’s interest in behavioral assessments related to the gut-brain axis in PFS. In fact, it has been previously demonstrated that this rat model develops depressive-like behavior following finasteride withdrawal, as discussed in the manuscript (DOI: 10.1016/j.psyneuen.2018.09.021). Furthermore, our ongoing studies suggest the presence of anxiety-like behavior and novelty avoidance in this model. These findings that we intend to submit as soon as possible further support the relevance of the model for studying neuropsychiatric phenotypes associated with PFS and reinforce the link between gut inflammation and behavioral alterations. This point has been clarified in the conclusions.
Round 2
Reviewer 1 Report
Comments and Suggestions for Authors
Dear Authors,
than you for the answers. Work has been improved and it could be published after minor revision.
Comments:
Fisher's post hoc test assume equal variance and is more prone to false assumption. Also box plot is used for better insight into the data with close median and mean value (since ANOVA requires normal distribution). Pleas explain have you checked is it Fisher suitable and is box plot better option from statistical point of view.
The dose of 1 mg/rat of allopregnanolone corresponds approximately to 3.3 mg/kg, assuming an average rat weight of 300 g.......is drug is sensitive than idea that rat with 250 g and the 350 g get same dose is not logical.
Author Response
1) We thank the Reviewer for the constructive comments and for the positive assessment of our work.
As previously noted, the uncorrected Fisher’s post hoc test was selected due to its simplicity and high sensitivity, particularly when dealing with a limited number of pairwise comparisons, where it maintains a reasonable balance between Type I error control and statistical power. In addition, the same statistical approach was employed in our previously published work (doi: 10.3390/biom12111567), ensuring methodological consistency and facilitating comparison across related studies.
We acknowledge that Fisher’s LSD test, like ANOVA, relies on assumptions regarding data distribution and variance. Prior to analysis, the distributional characteristics of our dataset were examined and found to be consistent with these requirements. No substantial deviations were observed that would compromise the validity of the applied parametric methods. Accordingly, the use of ANOVA followed by Fisher’s LSD was considered appropriate for the scope and structure of our data.
Concerning data visualization, bar graphs (showing mean ± standard error) were used instead of box plots, as this format more directly communicates differences in central tendency, which is the primary focus of ANOVA-based comparisons. While box plots can offer additional distributional insight, in the context of our study they were not deemed to provide substantial interpretive benefit beyond the information already conveyed.
In light of these considerations, the statistical and graphical approaches used were considered suitable for the objectives and scope of the present work.
2) Thank you for your comment regarding the dosing of allopregnanolone.
This dosing strategy was based on our previously published study in which the same allopregnanolone regimen proved effective (doi: 10.3390/biom12111567). Calculating the dose based on an average weight of approximately 300 g corresponds to about 3 mg/kg.
Although individual weight differences may exist, we believe they do not significantly impact the subchronic treatment outcome, as supported by the low standard deviations observed in our molecular results (doi: 10.3390/biom12111567; submitted article). These findings suggest that the treatment schedule remains effective despite small weight differences.
The treatment protocol consists of eight administrations on alternate days, over a total period of 16 days, which helps mitigate the impact of minor variations in the dose per kg received by each animal.
Furthermore, it is worth noting that in clinical practice, many drugs, such as finasteride, paracetamol, and aspirin, are often administered at fixed doses regardless of body weight, without compromising therapeutic efficacy. Therefore, we consider this approach appropriate and not a limitation of our study.
Considering these factors, we believe that a fixed dose per rat is appropriate and that weight variations do not significantly compromise the validity of the obtained results.